# Characterizing the Spatiotemporal Transcriptomic Response of the Right Ventricle to Acute Pressure Overload

**DOI:** 10.3390/ijms24119746

**Published:** 2023-06-05

**Authors:** Vitaly O. Kheyfets, Sushil Kumar, Paul M. Heerdt, Kenzo Ichimura, R. Dale Brown, Melissa Lucero, Ilham Essafri, Sarah Williams, Kurt R. Stenmark, Edda Spiekerkoetter

**Affiliations:** 1Paediatric Critical Care Medicine, Developmental Lung Biology and CVP Research Laboratories, School of Medicine, University of Colorado, Aurora, CO 80045, USA; 2Department of Anaesthesiology, Applied Hemodynamic, Yale School of Medicine, New Haven, CT 06510, USA; 3Vera Moulton Wall Center for Pulmonary Vascular Disease, Stanford University, Stanford, CA 94305, USA; 4Division of Pulmonary, Allergy and Critical Care Medicine, Stanford School of Medicine, Stanford University, Stanford, CA 94305, USA; 5Queensland Facility for Advanced Bioinformatics, The University of Queensland, Brisbane, QLD 4072, Australia

**Keywords:** right ventricle, transcriptomics, remodeling

## Abstract

This study analyzed microarray data of right ventricular (RV) tissue from rats exposed to pulmonary embolism to understand the initial dynamic transcriptional response to mechanical stress and compare it with experimental pulmonary hypertension (PH) models. The dataset included samples harvested from 55 rats at 11 different time points or RV locations. We performed principal component analysis (PCA) to explore clusters based on spatiotemporal gene expression. Relevant pathways were identified from fast gene set enrichment analysis using PCA coefficients. The RV transcriptomic signature was measured over several time points, ranging from hours to weeks after an acute increase in mechanical stress, and was found to be highly dependent on the severity of the initial insult. Pathways enriched in the RV outflow tracts of rats at 6 weeks after severe PE share many commonalities with experimental PH models, but the transcriptomic signature at the RV apex resembles control tissue. The severity of the initial pressure overload determines the trajectory of the transcriptomic response independent of the final afterload, but this depends on the location where the tissue is biopsied. Chronic RV pressure overload due to PH appears to progress toward similar transcriptomic endpoints.

## 1. Introduction

Cardiopulmonary disorders such as pulmonary hypertension (PH) and pulmonary embolism (PE) both lead to right ventricular (RV) dysfunction by increasing afterload and therefore triggering a cascade of remodeling events [1,2]. Although afterload reduction appears to show some therapeutic benefit for both etiologies [3], this is often not possible, and we don’t fully understand how the temporal sequence of remodeling is influenced by afterload severity or how this impacts recovery. In fact, our current understanding of the molecular mechanisms of RV remodeling and how it is coupled to structural changes and function is primarily limited to end-stage disease (with decompensated RV response) [4]. And while there has been a considerable amount of progress in characterizing the transcriptome and understanding the mechanisms behind RV remodeling [5,6,7,8,9], many unanswered questions remain. For example, (1) we have not characterized the spatiotemporal sequence of remodeling events and/or how this sequence is impacted by mechanical stress (e.g., should the transcriptome at a specific time point be correlated to the stress measured at that time point, or is it more influenced by mechanical stress at earlier stages?); and (2) we don’t fully understand how the remodeling phenotype is impacted by an acute (e.g., PE) vs. a progressive (e.g., PH) increase in afterload.

This study utilizes previously published data to answer these questions from a transcriptomic perspective. The objective(s) of this analysis were: (1) to assess how the spatiotemporal transcriptomic response is impacted by the severity of the acute overload insult (a surrogate for myocardial stress). (2) Compare pathways activated during an acute increase in RV afterload (e.g., rodent PE or pulmonary arterial banding models) to recently published studies that evaluated the RV transcriptomic response to a progressive increase in pressure overload induced by vascular remodeling (e.g., rodent models of PH [10]). Although this dataset has been previously analyzed by Watts, Zagorski, et al. [11,12,13,14], we utilized dimensionality reduction to analyze the entire spatiotemporal space across multiple loading conditions. This approach allowed us to visualize 11 conditions compared against each other, and to identify how the activation of critical remodeling pathways is impacted by time, mechanical stress, and region.

## 2. Results

### 2.1. How Is the Initial (First 18 h) Transcriptomic Signature of the RV Impacted by the Timing and Severity of Acute Pressure Overload?

PCA was performed to simultaneously visualize the temporal myocardial transcriptomic response to different levels of mechanical stress in one integrated analysis while overcoming the challenges of repeated pairwise comparisons for gene set enrichment analysis. Figure 1 shows how the RV remodeling trajectory is impacted by the severity of pressure overload in the initial phases of the acute insult. Figure 1a shows the temporal transcriptomic response of the high dose (HD) and low dose (LD) groups, with all data reduced using PCA to the PC1-PC2 plane, alongside the RVSP of both groups at each time point (see Figure 1b, curves reconstructed from ref. [13]). The data shows that the initial RVSP response to PE is significantly higher in the HD group, but the severity of pressure overload in both groups is equivalent at 18 h after PE. Nevertheless, the transcriptomic signature of both groups at early time points is almost identical, but the LD group appears to recover toward baseline by 18 h. However, the HD group (which experienced a significantly higher initial RVSP) revealed a transcriptomic signature at 18 h after PE that is considerably different from the LD group, even though both groups have the same RVSP at this time point.

Figure 1c reinforces the trends seen in Figure 1a by performing hierarchical clustering, viewed as a heatmap with a dendrogram, of the dataset shown in Figure 1a. This analysis also revealed that animals in the HD group cluster together at 18 h with a distinc transcriptomic signature from all other animals. LD animals at 18 h after PE —although also clustered together—are incompletely resolved from HD and LD rats at 6 h after PE.

Pairwise differential gene expression analysis allow one to identify differentially expressed genes between two conditions, and then rank those genes to study pathway activation. However, this approach carries multiple limitations when confronted with a high-dimensional experimental dataset with 11 conditions, such as the one considered here. Therefore, we were able to visualize the entire dataset and avoid pairwise comparisons using PCA, which allowed us to rank the genes for pathway analysis based on the PC coefficients. Given that the HD and LD animals at 18 h are most separated along PC1, we compared if the largest PC1 coefficients matched the most significant genes found from differential gene expression analysis between the two groups. Figure 1d,e shows that the top 20 genes (e.g., Il6, Il1r2, Il1rn, Ankrd2) with the highest influence on a point’s location along the PC1 axis (see Figure 1d) matched those from differential gene expression analysis between the HD and LD animals at 18 h after PE (see Figure 1e).

To assess how the temporal transcriptomic response is impacted by the severity of the acute overload insult, Figure 2 shows the temporal expression response of 12 genes in HD and LD rats during the acute response stage (0 to 18 h). These genes were chosen because they had the largest influence (corresponding to the largest PC coefficients) on PC1 and PC2, and we assume that the only stimuli that could have impacted RV gene expression were an acute elevation in mechanical stress. In other words, no chemical stimuli (e.g., Sugen, Monocrotaline) was administered. Therefore, genes that responded to this stimulus are those that directly or indirectly are stimulated by mechanical stress. However, because we did not have the information needed to compute RV myocardial stress for each rat (e.g., rat-specific RVSP, RV-free wall morphological measurements), we assume that it is directly proportional to RVSP in the first 18 h after PE, before any visible changes to RV free wall thickness.

Even though the highest difference in RVSP occurred within the first 2 h of PE (see Figure 1b), Figure 2 shows that 5 out of 6 top genes related to PC1 (Il1r2, Il6, Fcnb, S100a8, Ankrd2) appear to have a delayed response to an acute increase in mechanical stress, where expression is statistically similar between HD and LD rats at 2 or 6 h but then diverged to be significantly different by 18 h after PE (although the exact time point of divergence is not known). However, all genes that were important to PC2 appear to be completely unresponsive to the severity of myocardial stress (e.g., Tmem237, Dnajb5, Ubr4, Spata13, Twist2, Nova2) but are likely indirectly activated by elevated mechanical stress because -as previously mentioned- that was the only stimulus.

### 2.2. What Are the Pathways Associated with Acute RV Pressure Overload Induced by PE and How Do They Compare to Experimental Models of Pulmonary Hypertension (Sugen-Hypoxia and Monocrotaline)?

While rodent models of PH offer important insight into pulmonary vascular remodeling, they introduce confounding factors into the study of RV adaptation. Purely biomechanical models of RV pressure overload (e.g., pulmonary embolism [PE] or pulmonary arterial banding [PAB] models) allow researchers to focus on the impact of mechanical stress without needing to consider the impact of hypoxia or monocrotaline (MCT) on the RV transcriptome but do not replicate the progressive time-course of RV pressure overload in cardiopulmonary disease [15,16]. Therefore, the objective of this sub-section was to compare pathways activated in rat models of acute RV pressure overload vs. previously published [10] results from Sugen–Hypoxia (SuHx) and MCT models of pulmonary hypertension.

Figure 3a shows a biplot of all data (the data presented in Figure 1a combined with HD PE and CTL animals at 6 weeks) reduced to two dimensions: PC1 and PC2, which explain 46% of the data variance. Starting with the baseline points (t = 0 h), it appears the replicates at each time point are following a specific trajectory to t = 18 h. For data points to move along this trajectory, the activation of genes that correspond to the highest PC coefficients (β_i_, see Equations (1) and (2)) in PC1 and PC2 are the most important. However, the trajectory from t = 18 h to t = 6 weeks in RVOT samples travels vertically along PC2, thus suggesting that the PC2-responsive genes (and therefore, pathways) dominate this phase of the RV transcriptomic response. Surprisingly, the transcriptomic signature at the apex after 6 weeks of acute pressure overload resembles the transcriptional cluster of whole RV control tissue (t = 0 h). This result shows a spatially heterogeneous remodeling process within different regions in the RV.

For PC1, there are 768 genes that make up 50% of the cumulative sum of the PC1 coefficients. Out of those 768 genes, 493 are listed as being differentially expressed in the RV of MCT rats, but only 260 in SuHx rats [10]. For PC2, there are 712 genes that make up 50% of the cumulative sum of the PC2 coefficients. Out of those, 349 and 176 genes are common with the MCT and SuHx rats, respectively [10].

After ranking the genes according to their PC coefficients in PC1 and PC2, we performed gene set enrichment analysis to identify “hallmark” [17] pathways activated in the RV of HD PE rats. Gene set enrichment analysis is typically performed by ranking the genes according to their fold-change between some treatment group and control group (sometimes the *p*-value is also considered), so the use of PC coefficients in this work offered a slight adjustment to that approach. This allows us to create a list of the hallmark pathways (see Figure 3b,c) enriched when a data point moves along PC1 or PC2, respectively, and thus simultaneously compare against multiple experimental conditions.

A total of 27 pathways were significantly enriched in the RV of HD PE rats, with 10 common pathways between PC1 and PC2, 7 unique pathways in PC1, and 10 unique pathways in PC2. Even though these animals were exposed to an acute RV pressure overload, they shared 19 common pathways with Sugen–Hypoxia and MCT rats (at week 7) (compared to a pathways list from ref. [10]).

## 3. Discussion

In this study, we reanalyzed previously published data [11,12,13,14] (NIH/NCBI GEO database—GSE6104, GSE11851) to explore the spatiotemporal transcriptomic response of the RV to acute pressure overload in a rat PE model. The papers originally describing this dataset thoroughly illustrated the inflammatory response of the RV through neutrophilic infiltration and the presence of inflammatory markers but did not show how the transcriptome is related to the level of stress and/or illustrate the spatiotemporal nature of the entire transcriptomic response. Therefore, the objective(s) of our analysis were to: (1) describe the transcriptomic spatiotemporal trajectory of RV remodeling to an acute increase in pressure overload, and how this is impacted by the severity of myocardial stress; and (2) describe how acute changes in RV pressure overload—as seen in PE—compared against a progressive and cumulative increase in RV pressure—as seen in PH.

We utilized PCA via the PC1 vs. PC2 coefficients to visualize and quantify the spatiotemporal transcriptomic response of the RV to acute elevations in myocardial mechanical stress while simultaneously considering timing, pressure overload severity, and other spatial domains (e.g., apex vs. RVOT). The PC biplots can visually be inspected to evaluate all the conditions, including within and between condition variability, in a 2-dimensional plot. This approach is superior to traditional pairwise comparisons for datasets with many conditions, such as the one considered here because traditional differential gene expression analysis would require (for k = 11 conditions) $\frac{k\left(k-1\right)}{2}=$ 55 pairwise comparisons, yet without a comprehensive way to visualize the entire dataset. Similar to differential gene expression analysis, PCA coefficients can also be used to rank each gene’s contribution to the difference between multiple conditions, thus serving as input for gene set enrichment analysis. Using this analytic approach, we identified genes that were impacted by elevated mechanical stress (some likely indirectly), irrespective of the level of stress, that were not previously reported in papers analyzing the current dataset.

### 3.1. The Relationship between Acute RV Pressure Overload Severity and Temporal Transcriptional Trajectory

The data available for our analysis presented a unique opportunity to explore the synergistic impact of timing and severity, given that the LD and HD PE animals had drastically different initial RVSP levels (0 to 6 h after PE, see Figure 1b), but ultimately reached the same level at 18 h. Chronic RV remodeling generally consists of major changes to the internal structure [6,18], thus impacting tissue thickness and stiffness, which both heavily influence myocardial stress [19]. However, although a major infiltration of inflammatory cells was observed in these rats [13] and we do not fully understand how this impacted tissue structure, no changes in ECM structure or concentric hypertrophy were reported. Therefore, we suggest that comparing relative RVSP between multiple groups at early time points is analogous to comparing differences in myocardial stress. To this end, at 2 h after PE, the RV-free wall in HD animals was exposed to myocardial stress levels that were more than 20% greater than the LD animals, which set the two groups on drastically different remodeling trajectories. In fact, the severity of myocardial stress at the onset of PE (0 to 2 h) predetermined if the RV transcriptome would recover to baseline at 18 h after PE, even though RVSP (or myocardial stress) had reached the same level compared with baseline conditions by this point.

Focusing on the time course of the top 12 genes with the highest influence on a point’s location within the PC1-PC2 plane reinforced the idea that the initial severity of stress sets the trajectory of the response. Out of the 12 genes shown in Figure 2, it can be deduced that they all responded to mechanical stress because that was the only external stimulus acting on the tissue. The 6 genes most important for PC1 were all previously identified as differentially expressed at 18 h after PE [12]. These include inflammatory cytokines and their receptors (Il6, Il1r2), which are involved in many pathophysiological processes [20,21] and are known to be important regulators of myocardial response to stress/strain [22]. In fact, 5 out of the 6 genes are well known for their response to external stimuli [23] with Ankrd2 known to be critical for biomechanical stress response [24] and upregulated in SuHx rat RVs [25]. Interestingly, it took roughly 18 h for these genes to become upregulated in HD animals, relative to LD animals, even though the biggest difference in myocardial stress between these two groups occurred much earlier and had become mostly negligible by 18 h after PE. Although this finding is currently limited to the rodent PE model, it yields itself to the hypothesis—to be addressed in future studies- that the initial magnitude of stress might be the mechanism that explains why thrombolytic therapy is not equally effective across the PE risk stratification spectrum [26].

Our approach using PCA to uncover genes responding to acute mechanical stress/strain proved useful for those genes that revealed a temporary response. The genes found to be important to PC2 were not reported as differentially expressed in previous analysis of this data [12], which exposes a limitation of pairwise gene expression analysis when considering so many conditions. Clearly, these genes were responsive to mechanical stress in both the LD and HD groups, but their response appears to have been transient.

### 3.2. Acute vs. Gradual Increase in RV Pressure Overload: Can PAB or PE Rodent Models Teach Us about RV Response in PH?

In 2021, Park et al. [10] reported that the end-stage transcriptomic signatures between SuHx and MCT rats were moderately similar (2693 common genes; 34% of the differentially expressed MCT genes; 86% of the differentially expressed SuHx genes). The genes they identified as being important in RV remodeling were in broad agreement with previous RV gene-expression studies of both MCT [27,28] and SuHx [29] rats, and most of the genes found for MCT rats have also been reported as differentially expressed at the protein level [27].

SuHx and MCT have been used as common models of pulmonary arterial hypertension (PAH) to study the gradual pulmonary vascular remodeling that characterizes this cardiopulmonary disease [30], but their study confirmed that these two models generated roughly the same RVSP at end-stage disease with 25 common remodelings “hallmark” pathways significantly enriched (MCT and SuHx had 7 and 5 unique pathways, respectively) within the RV free wall. The PE rodent model considered in this study exposed the RV to an entirely different pressure overload trajectory and severity. While the Park et al. study simulated a gradual increase in RV pressure that rose to 90 mmHg, our study considered an acute increase in RVSP that initially (2 h after PE) rose to a maximum of 51 mmHg (in HD animals), but then gradually began to fall. Despite this difference, the PE rat model shared 19 out of the 25 hallmark pathways that were common in the SuHx and MCT models.

However, it is likely that the LD animals did not activate many of these pathways, or activated them temporarily, given that their transcriptomic signature returned to baseline at 18 h even though RVSP remained elevated, relative to baseline conditions. Therefore, given our previous work showing that the PAB mouse model of RV dysfunction closely replicates biomechanical changes measured in children with PAH [31], this study further supports the use of acute pulmonary constriction models to investigate RV remodeling and failure in cardiopulmonary disease. However, the severity of the initial insult is critical and future studies will need to compare the sequence of ECM remodeling (e.g., myocardial fiber re-orientation, stiffening, fibrosis [6,15,32]) and functional decline (previously shown to be closely related [33]) between these two models.

### 3.3. Spatiotemporal Transcriptomics at End-Stage Disease

There were three major limitations in the data available for the current analysis: (1) If we consider RV remodeling to be a sequence of events in 3 distinct stages: (a) onset, (b) progression, and (c) end-stage, then the current dataset does not offer any information about the progression stage; (2) transcriptomic analysis for the onset remodeling stage (0–18 h) was performed on the entire RV free wall, but data available for the end-stage (6 weeks) is localized to the apex and RVOT. Nevertheless, Figure 3 allows us to conclude that (1) the end-stage transcriptomic signature of the RV apex is more similar to control tissue than in the RVOT and (2) pathways unique to PC1 (e.g., apoptosis, protein secretion, adipogenesis) are activated at the onset and then deactivated during the progression stage of remodeling.

Given that it has become commonplace in some clinical centers to acquire cardiac tissue biopsies during corrective surgery, these findings highlight the importance of considering where and when that tissue should be acquired. The where and when question also needs to be a major point of discussion for future researchers interested in cardiac tissue response to mechanical stress, chemical stress, and/or the exploration of potential therapeutics.

A third limitation of the available dataset was that all data was collected in male rats, which prevented us from considering sex as a biological variable. This is likely an important limitation because of statistically significant differences in outcomes between men and women with PAH, which are likely—at least in part—related to RV response [34]. Future studies building on these findings will explore how the interaction between sex and mechanical stress impacts the transcriptomic, proteomic, and structural response.

## 4. Methods

### 4.1. Data Overview

The cDNA microarray data used for this study was obtained from a rat model of PE induced by the injection of polystyrene microspheres into the right jugular vein. Data were deposited at the NIH/NCBI GEO database (GSE6104, GSE11851), which consisted of Affymetrix Rat Genome 230 v2.0 microarray records (MAS 5.0 algorithm used for probe set-level normalization, scaling the trimmed-mean of each chip to a constant value = 500) for Sprague–Dawley male rats. All animals in the study were considered as part of the: (1) High dose group (HD)—2.0 million polystyrene microspheres per 100 g body weight were injected in the right jugular vein to produce severe PE; (2) low dose group (LD)—1.3 million polystyrene microspheres per 100 g body weight were injected in the right jugular vein to produce mild PE; and (3) control group (CTL)—received a 0.01% Tween 20 at 0.16 mL/100 g body weight vehicle, but not microspheres. Figure 4 shows an overview of the data available for the current study analysis. Additional details on animal surgery, tissue harvest, resulting hemodynamics, and microarray analysis can be found in ref. [11,12,13], and are omitted for brevity.

### 4.2. Data Analysis, Normalization, and Pre-Processing

All the data outlined in Figure 4 were combined for pre-processing and normalization. Starting with 31,099 probe IDs, the expressed sequence tags were removed to arrive at 15,651 probes. Confirmed batch effects between GSE6104 and GSE11851 were adjusted using the empirical Bayes method [35]. Our final analysis was performed on 5285 probes of interest, which revealed a significant difference in at least one experimental condition (see Section 4.4 Statistical Methods).

### 4.3. Dimensionality Reduction and Gene Set Enrichment Analysis

Those probes with a difference in normalized expression for at least one-time point were analyzed using principal component analysis (PCA) with each time point or location (e.g., RVOT) considered as a single observation and the probes as features. PC1 and PC2 are linear models of normalized probe intensities multiplied by PC coefficients ($\beta_{i}$, where *i* corresponds to the index of a single gene, see Equations (1) and (2)).
(1)$$PC1=\beta_{1,1}\left(Gene_{1}-\mu_{gene1}\right)+\beta_{2,1}\left(Gene_{2}-\mu_{gene2}\right)+\ldots +\beta_{4984,1}\left(Gene_{5285}-\mu_{gene5285}\right)$$
(2)$$PC2=\beta_{1,2}\left(Gene_{1}-\mu_{gene1}\right)+\beta_{2,2}\left(Gene_{2}-\mu_{gene2}\right)+\ldots +\beta_{4984,2}\left(Gene_{5285}-\mu_{gene5285}\right)$$

Therefore, the magnitude (rank) and sign (direction) of each coefficient dictate where each data point (or observation) is located in the PC1-PC2 plane. When multiple probes were mapped to the same gene, the overall rank (used for gene set enrichment, see below) and direction of that gene is found by summing the coefficients of each probe.

Gene set enrichment analysis (GSEA) was performed using the fast gene set enrichment package [36] (R 4.2.1) using the human Hallmark [17] molecular signature database (last updated August 2022). Rat gene Entrez IDs were matched to 4101 unique human Entrez IDs using the Ensembl database [37].

### 4.4. Statistical Methods

All microarray expression data were log2 transformed and normalized to the mean of the control group. Because of the relatively low sample size in each experimental condition (n = 5), all statistical comparisons were performed using a non-parametric Kruskal–Wallis test. A 2-tailed *p*-value was computed for each gene to identify probes of interest, with *p*-values calculated for repeated measurements adjusted using Benjamini and Hochberg correction to control for false discovery (false discovery rate, FDR < 0.01).

## 5. Conclusions

The PCA analysis, combined with pathway enrichment, performed in this study allowed us to reach 3 important conclusions about RV remodeling in response to an acute pressure overload: (1) the transcriptomic signature of RV tissue measured at a specific time point is likely more representative of the pressure overload severity at disease onset, rather than the pressure overload severity at tissue harvest; (2) performing transcriptomics on biopsied RV tissue that includes the RVOT and apex could potentially be minimizing the statistical effect of differentially expressed genes; and (3) acute RV pressure overload rodent models have both unique and some common pathways with PH models, which might assist with future study design depending on the research question. Future studies will aim to validate these results and directly link the transient transcriptomic signature to spatial differences in biomechanical stress, ECM remodeling, global tissue structure, and pumping function.

## Figures and Tables

**Figure 1 ijms-24-09746-f001:**
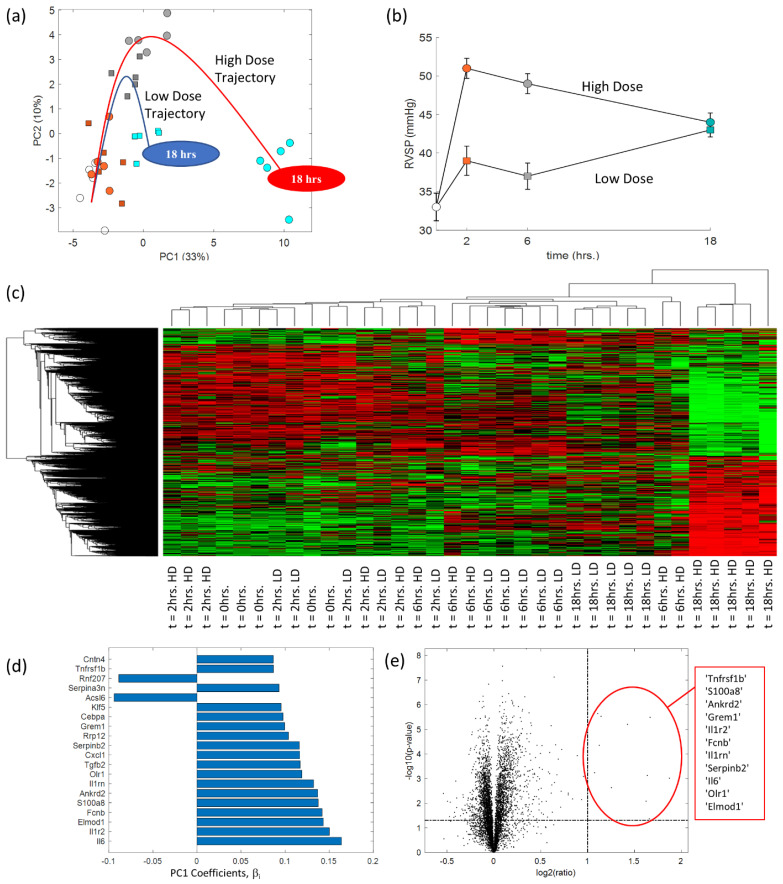
(**a**) Principal component analysis showing the transcriptomic trajectory (drawn by eye) in high dose and low dose rats. (**b**) RV systolic pressure (RVSP) measured in each experimental group. Data reconstructed from ref. [13]. Symbols and colours in (**a**,**b**) indicate dosing and time point, where each dot/square is color-coded based on the pressure condition and timing indicated in (**b**). (**c**) A heatmap and dendrogram showing hierarchical clustering of whole RV genes between low dose and high dose animals at 0 to 18 h after PE. (**d**) List of 20 genes that corresponded to the top 20 PC1 coefficients from (**a**). (**e**) Volcano plot of gene expression comparing low dose and high dose rats at 18 h after PE.

**Figure 2 ijms-24-09746-f002:**
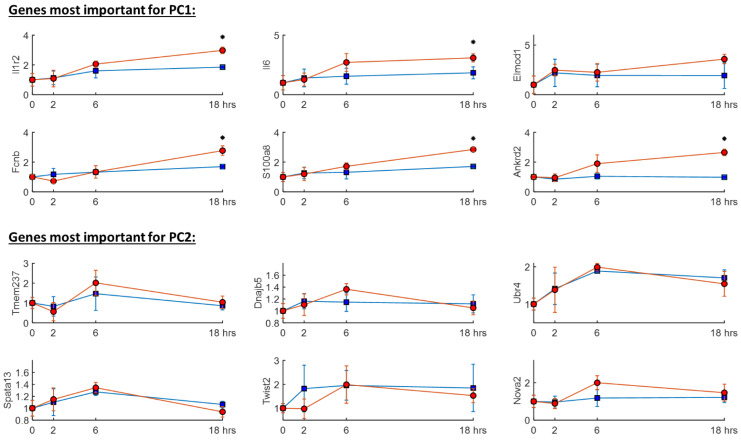
Time course plots of 12 genes, which are the most influential on PC1 and PC2, in HD (red circles) and LD (blue squares) animals. A *—symbol indicates statistical significance (P_2-tailed_ < 0.05 after a Benjamini & Hochberg adjustment for 36 comparisons) between LD and HD rats at a specific time point. Note: hrs = hours. The vertical axis for each gene is normalized to the mean of the expression at 0 h.

**Figure 3 ijms-24-09746-f003:**
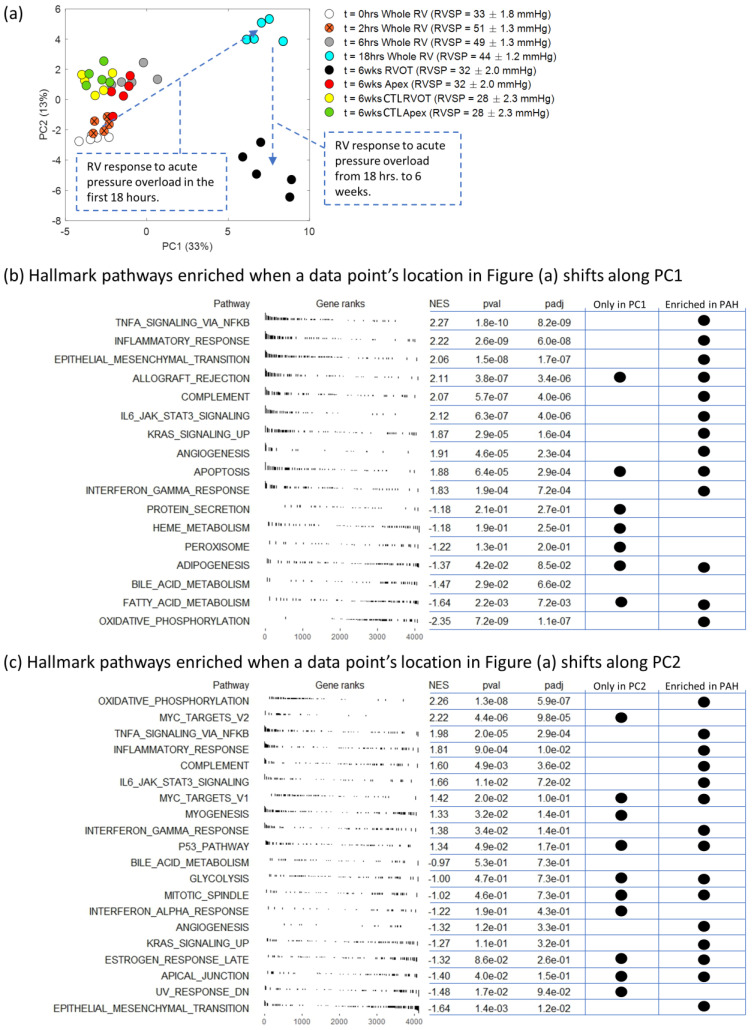
(**a**) PCA biplot showing the spatiotemporal response of the RV transcriptome to acute pressure overload in only HD rats, but with end-stage disease (6 weeks after PE) included into the analysis. (**b**,**c**) The hallmark pathways found by gene set enrichment analysis executed for genes ranked according to their contribution to PC1 (**b**) and PC2 (**c**). The final two columns indicate if that specific pathway is: (1) only enriched in one of the two PCs; and (2) also enriched in rat models of PAH. NES = normalized enrichment score; *p*-value is the a measure of the statistical significance of the enrichment score, and padj is the *p*-value after being adjusted for multiple comparisons using the Benjamini-Hochberg method. Within the legend of (**a**), RVOT = RV outflow tract; CTL = controls; RVSP = RV systolic pressure; hrs = hours; wks = weeks.

**Figure 4 ijms-24-09746-f004:**
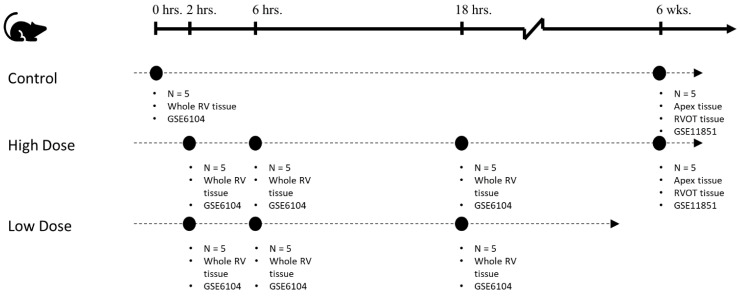
Overview of data analyzed for the current study, which included a total of 55 rat RV samples. At early time points (0–18 h) microarray analysis was performed on the entire RV. At later time points (6 weeks), the RVOT and Apex were isolated for microarray analysis. Note: hrs = hours; wks = weeks.

## Data Availability

The data used in this study is already publicly available. The data sources are given within the text.

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
