# Peer review of "Characterizing the Spatiotemporal Transcriptomic Response of the Right Ventricle to Acute Pressure Overload"

_ijms, 2023, doi:10.3390/ijms24119746_

Round 1
Reviewer 1 Report
1. I would like to know according to which criteria the amount of emboli was chosen for the HD and LD? Was there previous experimental models?
3. Have you use particular RV location
4. How could these results predict and guide the therapeutic management of PAH in humans
Author Response
We sincerely thank the editorial team and reviewers for their insightful comments and suggestions on our manuscript. We have carefully considered all reviewer remarks and have addressed them to the best of our ability.
- I would like to know according to which criteria the amount of emboli was chosen for the HD and LD? Were there previous experimental models?
Unfortunately, the paper outlining the original experiments [1] did not describe their reasoning for choosing the severity of pulmonary embolism applied to both groups. Interestingly, although not entirely surprisingly, it appears that HD mice has elevated RV systolic pressure (RVSP), relative to LD mice, immediately after surgery. However, RVSP in HD mice quickly began to decrease, while it increased in LD mice. Therefore, at 18-hours after surgery, RVSP in both groups were statistically indistinguishable. The reason for why the hemodynamics response in the two groups was so different is unclear, but it offered an interesting opportunity to compare the transcriptomic signature in both groups at 18-hours after surgery. Despite a similar RVSP in the two groups, the transcriptomic signature between the two groups was markedly different, which suggested to us that the transcriptomic signature at the post-remodeling phase is -at least in part- governed by the severity of the initial insult rather than the magnitude of myocardial stress at the point of tissue harvest.
- Have you used a particular RV location?
We admit that this was a limitation of the current study, but we were limited to the data that was available. The entire RV free wall was harvested and analyzed from animals sacrificed within the first 18-hours after pulmonary embolism. However, in rats where the tissue was harvested 6-weeks after pulmonary embolism, the RV was divided into the apex region and the outflow tract. Therefore, this prevented us from making a one-to-one comparison between the 18-hour HD animals and the 6-week HD animals. Nevertheless, we were still able to make two interesting observations from the 6-week HD animals: (1) the transcriptomic signature near the apex after 6-weeks of pressure overload was more similar to healthy tissue than to the RV outflow tract, which would suggest that this could be due to an inhomogeneous stress distribution along the RV free wall; (2) even though these animals were subjected to an acute (and severe) increase in RV pressure, their transcriptomic signature shared a lot of common genes and pathways with rodent models of pulmonary hypertension. Both points are important because they offer guidance on where to acquire tissue biopsies and -depending on the pathway of interest- when findings in an acute RV pressure overload rodent can be extrapolated to pulmonary hypertension research.
- How could these results predict and guide the therapeutic management of PAH in humans
This is a great question, but we were careful not to address this issue in the manuscript because it would require a lot of speculation. Although these findings would suggest that the location and timing of cardiac tissue biopsies should be carefully considered in a clinical setting, we feel the true value of our work lies in the identification of overlapping pathways between acute and PAH RV remodeling models. This offers a framework for what can be learned from an acute RV pressure overload model (e.g., pulmonary embolism, pulmonary arterial banding) about PAH RV remodeling and might reveal some common therapeutic targets.

Reviewer 2 Report
Pulmonary hypertension (PH) and pulmonary embolism (PE) both trigger a cascade of right ventricular (RV) remodeling events. The manuscript utilizes principal component analysis (PCA) and pathway enrichment to understand the mechanisms behind RV remodeling from the transcriptomic perspective on the published PE & PH data. The authors conclude that the severity of the initial RV pressure overload determines the trajectory of transcriptomic response, and chronic RV pressure overload seems to progress toward similar transcriptomic endpoints. The topic is of interest and significance.
However, I have some comments shown below.
1. The introduction could be expanded to provide a more comprehensive overview of the current state of research on RV remodeling and the transcriptome. This would provide context for the reader and help them understand the significance of the study in the manuscript.
2. Including a paragraph on statistical analysis in the methods section would help readers understand the statistical methods used to analyze the data and the significance of the results.
3. The authors could clarify the information presented in Figure 4b and the corresponding lines in the text to avoid confusing readers. Additionally, they should double-check the information on Page 22 to ensure it is clear and accurate. Fig. 4b, on Page 9, Line 178, says, “10 unique pathways in PC2.” Is it shown in Fig. 4b? Please double-check and clarify it. Also, check Line 251 “MCT and SuHx had 7 and 5 unique pathways.”
Page 22, please clarify the NES and its significance.
4. Discussion: I recommend discussing the pathway shown in Fig. 4 related to mechanical stress regarding RV remodeling.
Miner revision:
Ø “......ranging from hours to weeks after a 30 acute increase in mechanical stress.” The preposition and article “an” may be incorrect; Missing “from.”
Ø Please spells out when an abbreviation first shows up. For example, “RVOT” & “CO” in Fig. 4a; “PAB” on Page 7, Line 153; “CTL” on Page 8, Line 158; “MCT” on Page 8, Line 169, et al.
Ø Being consistent---For example, PH & PAH.
Round 2
Reviewer 2 Report
The manuscript is much improved.
I would recommend publishing after minor revisions, for example, the first paragraph in Introduction, increasing the indent, and being consistent with other sections.
Statistical Methods:
“All microarray expression data were log2 transformed ......”
“......with p-values calculated by repeated measurements......”
Author Response
We thank the reviewer for carefully going through the second version of our manuscript. We believe the concerns were addressed in this revised submission.
I would recommend publishing after minor revisions, for example, the first paragraph in Introduction, increasing the indent, and being consistent with other sections.
We fixed this and went through the manuscript to ensure consistency.
Statistical Methods:
“All microarray expression data were log2 transformed ......” This has been fixed.
“......with p-values calculated by repeated measurements......” This has been fixed.